# Real-time simultaneous refractive index and thickness mapping of sub-cellular biology at the diffraction limit

Arturo Burguete-Lopez [1], Maksim Makarenko[1], Marcella Bonifazi[1,2], Barbara Nicoly Menezes de Oliveira [1], Fedor Getman[1], Yi Tian[1], Valerio Mazzone[1,2], Ning Li[1], Alessandro Giammona[3,4], Carlo Liberale [3] & Andrea Fratalocchi [1✉]

Mapping the cellular refractive index (RI) is a central task for research involving the composition of microorganisms and the development of models providing automated medical screenings with accuracy beyond 95%. These models require significantly enhancing the state-of-the-art RI mapping capabilities to provide large amounts of accurate RI data at high throughput. Here, we present a machine-learning-based technique that obtains a biological specimen's real-time RI and thickness maps from a single image acquired with a conventional color camera. This technology leverages a suitably engineered nanostructured membrane that stretches a biological analyte over its surface and absorbs transmitted light, generating complex reflection spectra from each sample point. The technique does not need pre-existing sample knowledge. It achieves $10^{-4}$ RI sensitivity and sub-nanometer thickness resolution on diffraction-limited spatial areas. We illustrate practical application by performing sub-cellular segmentation of HCT-116 colorectal cancer cells, obtaining complete three-dimensional reconstruction of the cellular regions with a characteristic length of 30 $\mu$m. These results can facilitate the development of real-time label-free technologies for biomedical studies on microscopic multicellular dynamics.

[1] PRIMALIGHT, Computer, Electrical and Mathematical Sciences and Engineering (CEMSE), King Abdullah University of Science and Technology (KAUST), Thuwal 23955-6900, Saudi Arabia. [2] Physik-Institut, University of Zurich, Winterthurerstrasse 190, Zurich 8057, Switzerland. [3] Biological and Environmental Science and Engineering Division (BESE), King Abdullah University of Science and Technology (KAUST), Thuwal 23955-6900, Saudi Arabia. [4] Institute of Molecular Bioimaging and Physiology (IBFM), National Research Council (CNR), Segrate, Italy. ✉email: andrea.fratalocchi@kaust.edu.sa

The study of cellular refractive index (RI) attracts broad interest as it enables a fundamental understanding of cellular growth, differentiation, and mutations[1–4], the development of pharmaceuticals[5,6], and the identification of severe diseases[7,8], including early signs of cancer[9,10]. Traditional research focused on mapping the refractive index to elementary observables such as density or protein concentration[11–13]. More recent pioneering works combine RI measurements and machine learning (ML) to develop automated bio-imaging workflows for label-free imaging and medical diagnosis[14–16].

This trend exploits the ability of artificial neural networks to identify sparse patterns encoded in RI raw data, learning key biological traits for enhanced screening applications[17]. Machine learning models trained on RI data reported accuracy as high as 95% in the classification of cells, showing great potential in stain-free digital pathology and drug development[18]. Similarly, RI-trained deep learning models demonstrated the ability to label unstained cells digitally, enabling non-invasive analysis of sensitive specimens for in vitro fertilization[19].

Today, progress in this field requires developing technologies capable of producing training data with high-throughput[8,16,17]. Currently, state-of-the-art RI mapping compromises between accuracy and speed. The most accurate RI mapping method, plasmonic probe scanning, provides RI measurements with $10^{-5}$RIU resolution, but requires tens of minutes to scan a single line of a sample's RI map[20]. Single-shot quantitative phase microscopy (QPM) provides faster estimations but requires pre-existing knowledge of the cell thickness[21]. However, the unavoidable uncertainty in estimating the cell geometry is a source of significant errors in QPM-based RI estimation, leading to measurements that vary between $10^{-2}$RIU and $10^{-4}$RIU in resolution [22–25].

A technology providing fast and accurate RI mapping in real-time while enhancing present methods could also open new opportunities, including the real-time study of pharmacological effects on cells or the subcellular dynamics occurring during pathological bacteria replication[26].

In this work, we implement a machine learning technique that recovers point-to-point and segments in real-time a biological specimen's RI and thickness. This approach employs a single image collected by a conventional color camera and a suitably engineered nanostructured membrane. It does not require pre-existing knowledge of the sample. We demonstrate sub-cellular segmentation in real-time without staining or manual labeling using an off-the-shelf digital camera and a traditional bright-field microscope.

## Results

Figure 1 illustrates the main idea of the proposed RI measuring technology. It leverages a suitably engineered ultra-dark hydrophilic surface of palladium (Pd). When a specimen carried inside a droplet of phosphate buffer solution (PBS) deposits on the Pd surface, it anchors itself to the surface at multiple points. The hydrophilic nature of the Pd surface causes the PBS to spread over the sample, resulting in the evaporation of the liquid within one minute of the deposition. The lack of liquid produces progressive dehydration of the specimen, causing it to flatten and stretch on the surface, forming a suspended, thin biological film. When a white light source illuminates this structure, the reflection spectrum shows complex frequency modulations based on interference-generated structural colors (Fig. 1b). A conventional red, green, and blue (RGB) camera converts every pixel's input spectral power distribution (SPD) into a triplet of RGB values.

The camera integrates its color-matching functions (CMFs) with the input SPD during the conversion. The CMFs (Fig. 1c, $b(\lambda)$, $g(\lambda)$, and $r(\lambda)$ curves) represent the device's sensitivity to the three primary color bands. The output RGB value encodes unique information on the biological properties of the analyte, such as its thickness and refractive index. After imaging, machine learning software performs a pixel-by-pixel segmentation by recovering the thickness and refractive indices from the RGB features encoded by the camera. Figure 1d shows an example three-dimensional reconstruction of the thickness map of an HCT-116 colorectal cancer cell. The layered colors on the panels of Fig. 1e highlight distinct sub-cellular clustered structures with similar refractive indexes. This approach does not rely on cell preparation and is free from chemical alterations. At the same time, it enables automated measurement of the thickness and refractive index information in a single parallel acquisition with diffraction-limited spatial resolution. This technique requires only a conventional camera and a reflection microscope, opening up the possibility of in-situ integrated setups compatible with equipment for cell culture growth and development studies.

Figure 2a shows an example of the experimentally fabricated Pd surface used for the analysis. Surface manufacturing uses electro-deposition of Pd on a gold-coated glass piece (more details in Methods). We optimize the deposition potential and time to create large and prominent tree-like features (Fig. 2a, black area) and achieve broadband light absorption. The combination of the Pd surface texture and its low reflectivity produce the cell stretching to thin film effect while simultaneously allowing the thin film interference colors to be detectable. Figure 2b, d shows scanning electron microscope (SEM) images obtained from the top and cross sections of the sample. The deposited Pd grows on a layer approximately 30 $\mu$m in height and comprises irregularly shaped pillars, producing a pattern reminiscent of a rainforest canopy. The insets in panels b and d show that each pillar is further textured at the nanometer scale, contributing to their hydrophilic nature. Figure 2c shows a photograph of the Pd surface at ×100 magnification under a brightfield reflection microscope. The image highlights the highly absorbing nature of the sample, with only minor light reflection at the tips of the Pd pillars under direct Köhler illumination. The inset in Fig. 2c reports the region's reflectivity across visible wavelengths measured with an integrating sphere, showing that the nanostructured Pd reflects less than 2% of visible light relative to a silver mirror. Most of the light scattering from the pillars occurs at high angles, enabling the detection of thin film interference components that scatter within the numerical aperture of the microscope objective.

The RGB color features of a stretched biological specimen depend on its local thickness and refractive index uniquely. Figure 3a illustrates this point quantitatively. The figure presents examples of standard RGB (sRGB) colors generated via thin-film interference at four representative film thicknesses (see Methods for more detail). For each thickness, the refractive index varies in the biological range from 1.33 to 1.55. Figure 3a shows that sRGB features encode unique combinations of thickness and refractive index that do not intersect, thus permitting the retrieval of these quantities with no ambiguity. This feature allows for overcoming the limitation of QPM methods, which require pre-existing knowledge of the sample thickness.

Figure 3b, c present a theoretical analysis of the resolution limits of this method. The $y$-axis of the plots represents the level of variation, in units of bits, that the image file may suffer from due to thermal, electrical, or illumination fluctuations in the experimental setup. This value can be estimated by examining the variation in pixel values between images of the same object taken at different times. For a given bit variation, each circle marks the thickness or refractive index resolution below which two distinct biological structures yield the same RGB triplet. The dotted lines of the image help visualizing the resolution dependence on bit depth, but the plots are not continuous as a discrete variation of camera bit depth yields a discrete variation in the sensitivity of the technique.

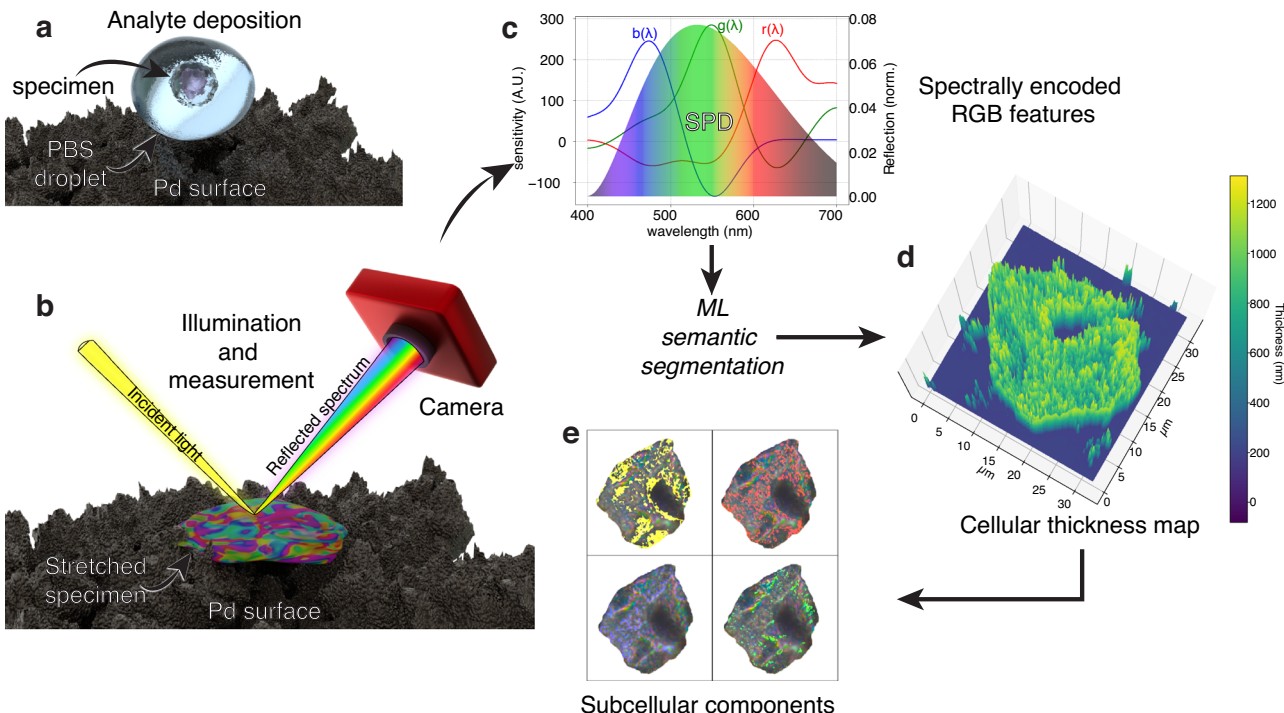

**Fig. 1 Technique overview. a** Deposition of a biological specimen using a PBS droplet onto a nanostructured Pd surface. **b** Stretched specimen acting as a thin film that exhibits interference-based colors when illuminated. Recording of spatially dependent colors by a digital camera. **c** Camera conversion of analyte SPD into RGB values. **d** Recovered thickness map for an HCT-116 colon cancer cell. **e** Micrographs of an HCT-116 cell. The color overlays indicate subcellular regions with similar refractive index.

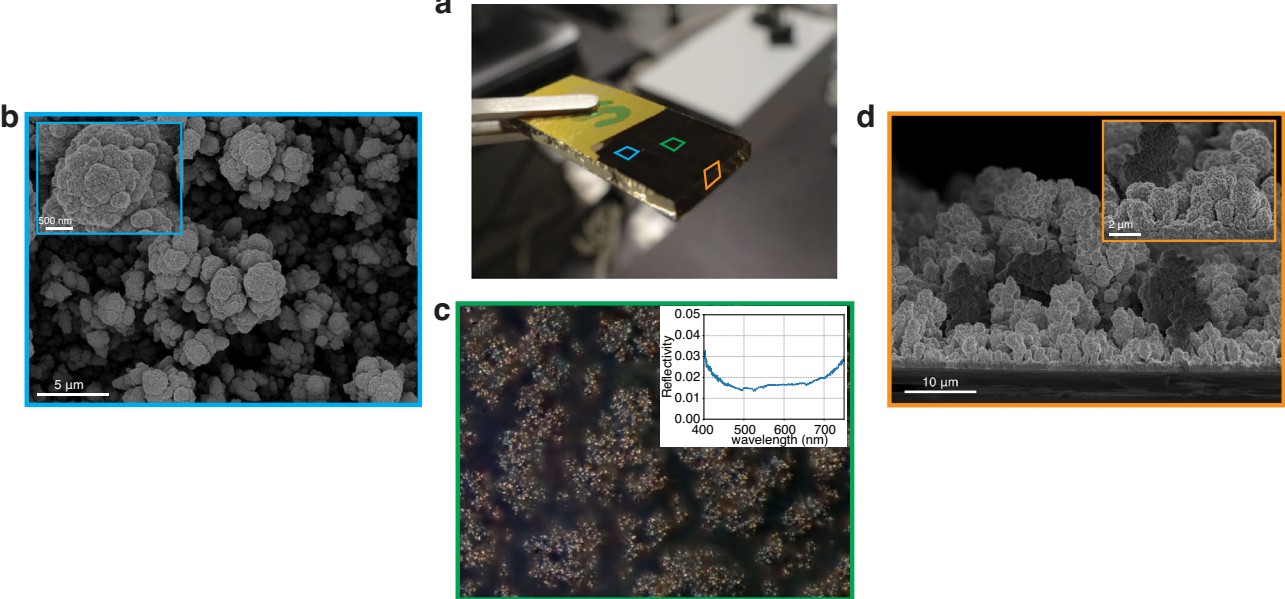

**Fig. 2 Cell–substrate interaction. a** Photograph of nanostructured Pd sample. The color squares correspond to the regions imaged in (**b–d**). **b** Overhead SEM micrograph of nanostructured Pd. **c** Optical micrograph of nanostructured Pd. The inset shows the reflection average reflection spectra of the area. **d** Cross-sectional SEM micrograph of nanostructured Pd.

Figure 3b, c shows that this technique achieves state-of-the-art refractive index resolutions ($10^{-4}$) for a 16 bit per color channel camera. Likewise, this method reaches nanometer thickness resolution when employing cameras of 14 bits per channel or higher.

While the mapping between a spectrum and an RGB triplet is unique within the expected biological thickness and RI ranges, in a limited number of cases, the conversion of an SPD to the bit-limited RGB space of the camera yields very close RGB values, a phenomenon known as metamerism. Figure 4a shows an example of this by plotting the theoretical reflection spectra of two metameric films, S1 and S2. The two spectral curves represent the response of thin films deposited over a silicon substrate with RI values of 1.41, 1.49 and thickness values of 588 nm and 356 nm, respectively. These thicknesses and RI values lie within the

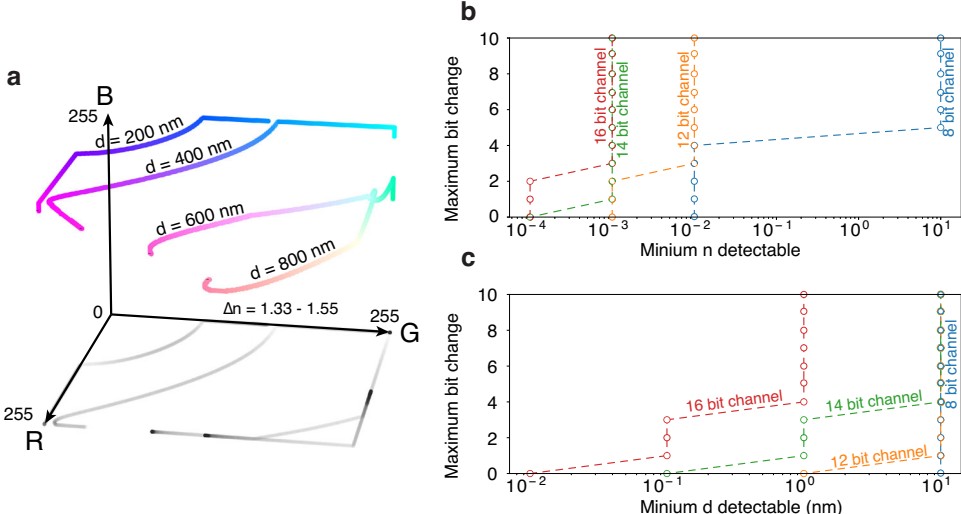

**Fig. 3 Theoretical performance. a** Biological thin film colors in the sRGB colorspace for four different thicknesses as the refractive index varies from 1.33 to 1.55. **b, c** Sensitivity limits for refractive index and thickness values recovery as a function of the channel bit depth of the camera used and the stability of the image values. The plot is composed of discrete points with the dashed lines intended to help visualizing the trends.

**Fig. 4 Machine learning RI and thickness algorithm. a** Reflection spectra and RGB color of metameric thin films S1 and S2. **b** Clustering of thin film sample into two pixel groups. **c** Cost maps for four pixels of cluster 1. **d** Expanded view of the cost map of pixel ii, the pink and blue areas indicate the probability of the thickness and RI values respectively. **e** Pooled cost function for the pixels of cluster 1.

expected range of biological specimens[27]. While the two films have different properties, when integrated through an 8-bit camera's CMFs they map to RGB colors that are almost indistinguishable to the human eye: RGB = [149,251,122] (S1) and RGB = [141,251,134] (S2). We designed and implemented a machine learning recovery procedure that retrieves thickness and RI without human bias or intervention for these challenging metameric scenarios.

The process starts by accurately characterizing the camera's CMFs through supervised learning. In this step, we used a training and validation experimental dataset of 65 thin films of known thickness and RI. We manufactured these thin films via the spin coating of PMMA photoresist on silicon wafer pieces at different speeds and measured their thickness and RI through spectroscopic ellipsometry (see Supplementary Figs. 1 and 2a–d). We then acquired reflection spectra and photograph pairs for each film sample. Using these samples, we trained a regression model using non-linear basis functions (see Supplementary Note 1 for implementation details). This approach yields the CMFs up to the desired resolution in frequency, controlled by the size of the regression model. This training process allows the measurement of any biological thin film imaged by the camera, as the ML algorithm is agnostic to the type of cell or imaged material, learning only the relation between the spectral power distribution of the specimen and the color outputted by the camera.

After estimating the CMFs, the ML recovery algorithm can extract the thickness and RI values for each pixel of a sample's image. However, due to metamerism, working with each pixel as an isolated element can result in incorrect recoveries. The ML algorithm addresses this by pooling information from pixels with close RGB values, generating groups of adjacent pixels possessing similar RGB colors in the image. This process uses an unsupervised k-means clustering algorithm that labels pixels of similar RGB colors as belonging to the same cluster. The ML recovery procedure automatically sets the number of clusters to yield an average variation of less than 2% between the RGB values of the pixels in each cluster and the cluster centroid RGB value. We set this value as a threshold found through successive iterations of the algorithm, with the condition that a lower value would result in the differences in recovered RI and thickness values for the pixels in a cluster being below the sensitivity of our setup. Slight RGB differences between adjacent pixels correspond to nanometer scale fluctuations in the material's thickness, which the camera perceives even at the single nanometer. (see Fig. 3c).

Figure 4b illustrates clustering for an experimental thin film sample manufactured with the parameters of S2. Running the clustering process results in two clusters for the image, one corresponding to the green area of the thin film and another for the black edge of the field stop of the microscope used to take the image. The average difference between the RGB triplets in the green cluster and the centroid RGB value is 0.86%.

In each cluster, ML recovery employs a pooling strategy similar to using pooling layers in convolutional neural networks[28]. For a subset of 1000 randomly sampled pixels within the cluster, we compute a mean square error (MSE) cost map:

$$\text{MSE} = \frac{1}{3}\left|\mathbf{X} - \hat{\mathbf{X}}\right|^2 = \frac{1}{3}\sum_i \left(X_i - \hat{X}_i\right)^2, \quad (1)$$

where $\mathbf{X} = [X_1, X_2, X_3] = [R, G, B]$ is the measured RGB triplet of the pixel, and $\hat{\mathbf{X}} = [\hat{X}_1, \hat{X}_2, \hat{X}_3] = [\hat{R}, \hat{G}, \hat{B}]$ a numerically computed thin film RGB value from a table of RGB values corresponding to thin films of known RI and thickness values (see Supplementary Fig. 2e, f). We calculate the RGB table only once, and the cost map executes in parallel for each cluster. Figure 4c illustrates the cost maps associated with four random pixels in the cluster, and Fig. 4d presents an expanded view of the map of pixel ii. Because of metamerism, the MSE cost map shows two local minima (yellow areas), one corresponding to the thickness and RI values of S1 and the other to the values of S2. The ML recovery procedure computes the probability of each of these RI and the correct thickness values by slicing the MSE map along each axis and comparing the minimum values (Fig. 4d pink and light blue probability areas). This step results in a 0.62 probability that the acquired RGB value belongs to the RI and thickness of S1 for pixel ii.

The algorithm then pools together the cost maps of each pixel within the same cluster to improve the low-confidence probabilities and correctly identify the thickness and RI values of the film. This procedure averages out outliers and yields the MSE map depicted in Fig. 4e. This map presents a single minimum, which correctly corresponds to the sample's thickness and RI values with unitary confidence and no ambiguity.

Figure 5 summarizes validation results for the ML RI and thickness recovery on synthetic cell-like objects with engineered thickness and refractive index. These synthetic cells are ≈30 μm wide squares of cured SU-8 photoresist (see Methods for fabrication details). We measured the cell's thickness $t$ using optical and contact profilometry (see Supplementary Fig. 1), obtaining $t = (567 \pm 6)$ nm, and obtained the ground truth RI from the resist manufacturer datasheet. Figure 5a shows a photograph of a synthetic cell through a reflection microscope at ×100 magnification. The blurring on the right side of Fig. 5a does not originate from a thickness variation but is the result of a slight tilt of the cell, which places this area outside the depth of field of the ×100, 0.9 NA, objective we use to acquire the image. The cell is of a near uniform green color except for two dark spots within its area, which correspond to supporting Pd pillars seen through the cell. Figure 5b presents a three-dimensional image of the cell positioned on the Pd substrate, illustrating how the cell is supported at a slight angle by these two pillars. Figure 5c, d shows the ML calculated thickness and RI maps of the artificial cell structure. As the cell is uniform in both thickness and refractive index, the plots present constant values for both quantities over the cell's surface, save for the areas where the Pd pillars are detected. Our algorithm treats the Pd pillars background as a black thin film during the calculations, and will not further processes these areas for RI and thickness recovery. Figure 5e, f presents the absolute uncertainty against the ground truth values. We calculated the uncertainty as the difference between the values recovered by our algorithm and ground truth measurements of the refractive index and thickness. The procedure yields results with an average discrepancy of 0.6 nm in the thickness recovery compared to the average cell thickness obtained with the profilometer measurements and of $3 \times 10^{-3}$RIU compared to the datasheet RI over the synthetic cell area.

Figure 6 presents the results of the recovery process applied to a natural cell. Figure 6a shows a photograph of an HCT-116 colon cancer cell after deposition and stretching on the Pd surface. Spatially varying thin film interference colors are visible across the specimen. The dark spots in the central part of the cell correspond to debris from a Pd pillar that moved over the cell during the deposition process. The blurriness on edge results from the short depth of field of the ×100, 0.9 NA objective used to capture the image. We set the microscope to focus on the largest possible cell area as the sample must be in focus to prevent overlap between neighboring pixels' RGB values and allow the technique to obtain sharp RI and thickness maps. Figure 6b, c shows the ML computed RI and thickness maps of the specimen using 50 color clusters. This number results in a maximum variation considering all clusters of 1.98% between the RGB values of the pixels and their cluster centroid RGB triplet. Consistently with previously reported RI maps for HCT-116 cells, no sharp nucleous-cytoplasm boundary is apparent, however, the RI values shown in Fig. 6b are larger than those reported in the literature for living HCT-116 cells by approximately 0.1 RIU[29,30]. This RI increase is a consequence of cell dehydration, and is consistent with the previously reported RI increase of up to 0.15 RIU across the visible wavelength range for dehydrated tissues and isolated cells undergoing dehydration[31,32]. The ML algorithm correctly isolates the Pd background in both results, grouping all pixels with low RGB values into the background cluster. This clustering step produces a sharp boundary separating the cell from the Pd

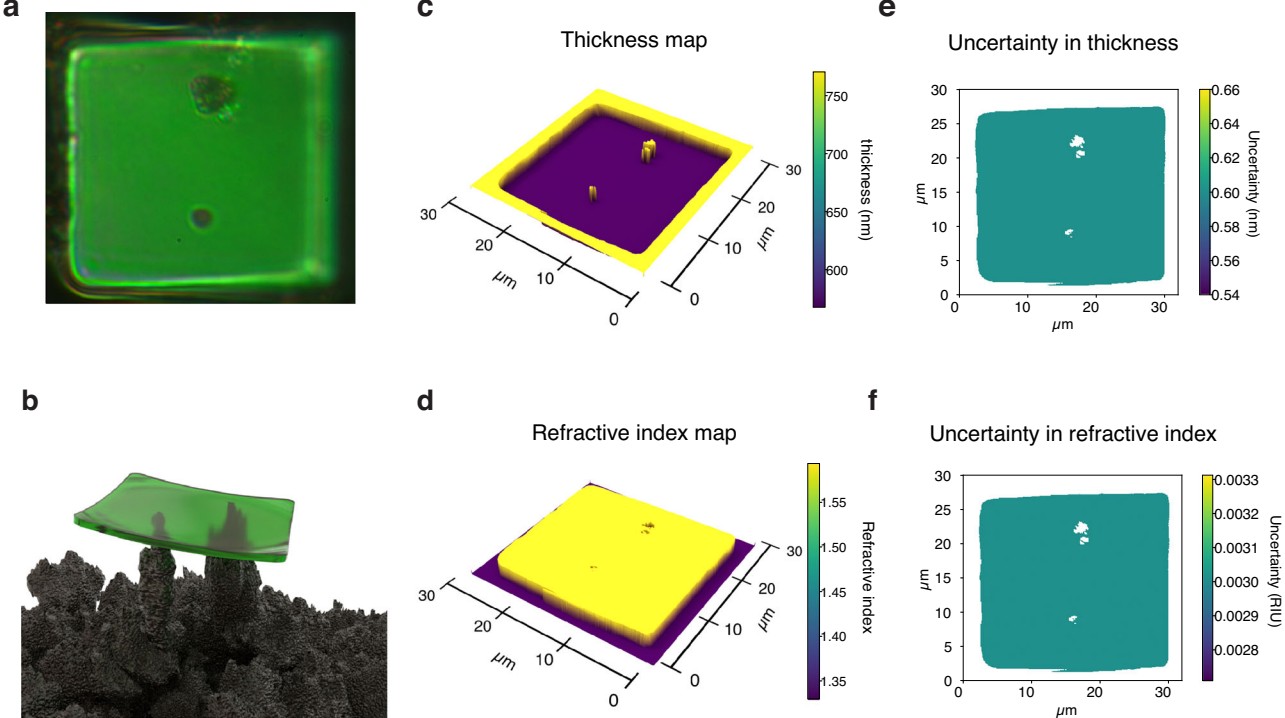

**Fig. 5 Validation with synthetic cell. a** Photograph of a synthetic cell as seen under ×100 magnification on top of the Pd substrate. The two dark spots correspond to Pd pillars visible through the cell. **b** 3D model showing the relative positioning of the synthetic cell on the Pd pillars. **c, d** 3D reconstruction of the thickness and refractive index maps obtained for the synthetic cell. **e, f** Uncertainty maps for the thickness and RI of the synthetic cell.

according to whether the RGB values of the pixels are above the threshold the algorithm defines as the background. The algorithm likewise identifies and groups the Pd debris on the cell with the background pixels. Figure 6d illustrates the ten most significant clusters, excluding the background, that the algorithm finds for the photographed cells. The cell's dark gray interior represents the remaining smaller clusters. Each cluster corresponds to groups of pixels the algorithm identifies as having equal RI and thickness values. Figure 6e is an SEM close-up of the specimen. The panel shows the thin film nature of the cell and the raised height of the specimen edges relative to the rest of the body that cause the edge blurriness of Fig. 6a. We ensured the SEM imaged cell was the same as the cell shown in Fig. 6a by scratching markings in the Pd surrounding the cell. We estimated the cell thickness from the SEM image by measuring the number of pixels in the image corresponding to the raised border of the cell, and then multiplying this value by the size in nanometers of one pixel. The estimated cell's thickness from the SEM image lies between 250 nm and 800 nm, in good agreement with reconstructed values in Fig. 6c. Figure 6f presents a complete 3D reconstruction of the cell thickness profile with a color overlay that varies according to the point-to-point RI value.

## Discussion

This work introduced a machine-learning-based single measurement platform for recovering thickness and refractive index maps from cells. This technology uses a conventional color camera and a nanostructured surface that stretches analytes while absorbing transmitted light. The technique provides nanometer accuracy for thickness measurements and up to $10^{-4}$ RIU for RI measurements. We validated this approach by showing real-time data acquisition and measurement of cellular and cellular-like structures.

The proposed platform enhances the capabilities of existing systems, providing a single automated platform yielding accurate

predictions for the real-time processing of biological data. Measurements leverage standard laboratory equipment in the form of an epi-illumination microscope, whose footprint can, in the future, be reduced to a single flat-optical surface, providing a high-level of integration[33]. The system requires only minimal preparation of the cells and employs industrial electrochemical manufacturing that scales up to any desired area.

Beyond the initial demonstration and proof-of-concept, this work opens multiple future research avenues. At the software level, new image routines can automate locating areas where the cell presents folding effects likely to lead to spurious results and detect out-of-focus regions for their exclusion from the analysis. Work in this area can leverage the large body of research developed in video understanding using deep-learning[34–36].

From a more material-oriented perspective, further study can investigate alternative nanostructuring configurations to stretch different cell classes better while minimizing light transmission. Varying the sample illumination by introducing filters on the microscope light path can also lead to further research to improve detection accuracy for a given camera bit conversion precision.

While the cell measurement is instantaneous in this technique, combining this measuring platform with automated pipetting systems could enhance the setup speed by accelerating the specimen deposition process. Instruments that accelerate the dehydration process by positioning the substrate in a dehydration chamber would further reduce the setup to measurement time, providing options for developing commercial cytometry equipment based on this platform.

The proposed technique could help biological investigations with a real-time and inexpensive platform for various applications to automate refractive index measurements. While an exhaustive analysis goes beyond the scope of this work, we discuss a few relevant examples next. In the cancer care community, this method could help by automatically screening refractive index

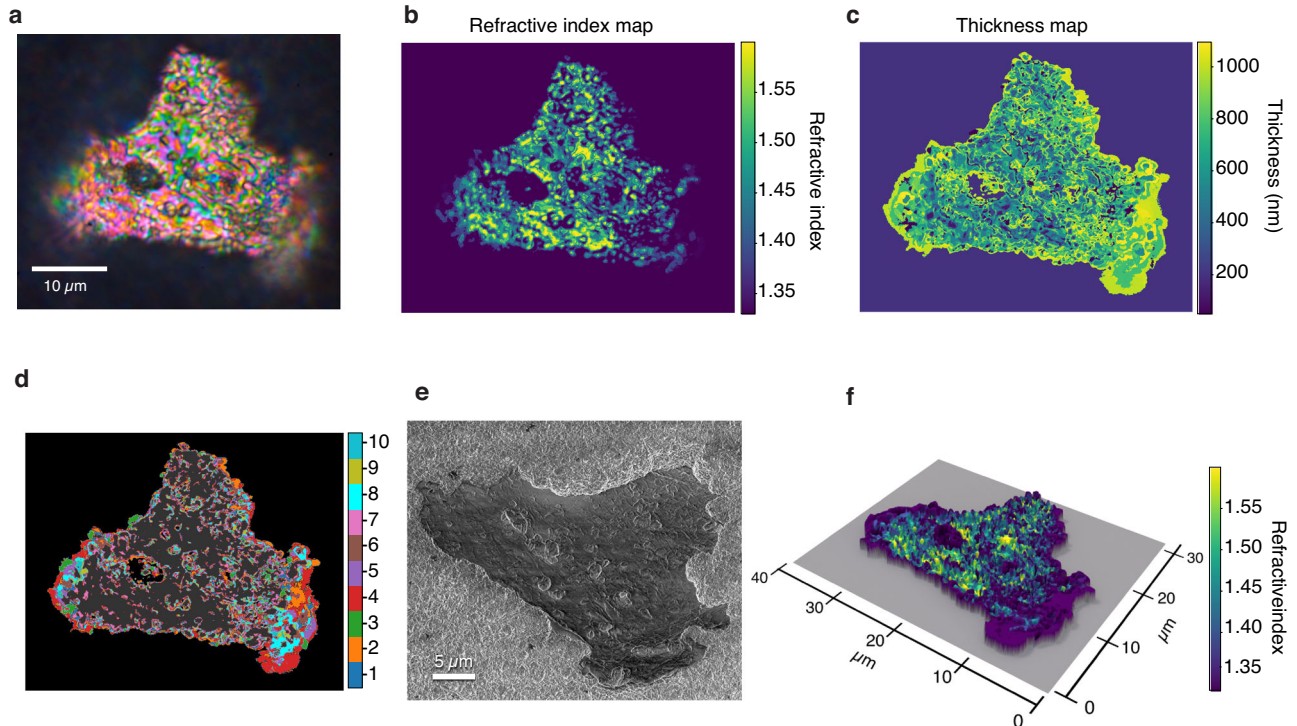

**Fig. 6 3D reconstruction and segmentation of an HCT-116 cell. a** Photograph of an HCT-116 cell stretched on the Pd substrate showing thin film interference based spatially dependent colors. **b**, **c** ML recovery results for the thickness and RI of the specimen in (**a**). **d** Ten largest clusters found for the cell depicted in (**a**), the remaining clusters are grouped as the dark gray interior of the cell. **e** SEM micrograph of the cell on the Pd substrate. **f** 3D reconstruction of the thickness map of the cell with overlayed RI information.

markers in cells with simple, integrated equipment with massive deployment. Differentiating between healthy and cancerous cells can be achieved with a RI resolution of $10^{-2}$RIU[25,37], which we demonstrate in this work.

Future research could use the method introduced here to study in-vitro cell culture sheets. Placing cellular monolayers on the Pd substrate would enable the single measurement study of cellular growth, differentiation, and responses to biophysical or biochemical cues through the lens of RI mapping for multicellular assemblies[1,38]. The technology can also help the detection of other refractive index markers indicating changing diseases in the human blood, such as malaria, with a resolution of $10^{-2}$RIU in the RI map of erythrocytes[7]. The result of this work can also benefit pharmacological research. It could provide low-cost and easy-to-use equipment that monitors the effect of new antibiotics from changes in the refractive index of bacterially infected cells, a task that requires a $10^{-3}$RIU resolution[39], attainable with this technology.

## Methods

**Nanostructured Pd sample fabrication**. Pd samples were fabricated through an electrodeposition process using a three electrodes configuration with an Autolab PGSTAT204 potentiostat. The working electrode was connected to a gold coated glass substrate piece (EMF corporation) cut to dimensions of 1 cm × 2.54 cm. The counter electrode was connected to a 1 cm$^2$ platinum sheet electrode, and a Ag/AgCl electrode was used as the reference. Electrodeposition was performed by submersing half of the coated glass substrate into a 40 mmol $K_2PdCl_4$ in 0.5 mol HCL solution, and applying a −250 mV bias for 300 s.

**Computation of thin film colors**. The RGB colors of thin films were computed by using a python implementation of the transfer matrix method[40]. For a given thickness and refractive index the

reflection spectra was obtained by computing the reflected wave power for normally incident light on a single layer of material surrounded by air from 400 nm to 800 nm in steps of 1 nm. Numerical integration of the reflection spectra with each of the CIE 1931 2° observer XYZ color matching functions was then carried out to obtain the tristimulus values. These values were converted to a normalized linear RGB space by matrix multiplication. RGB triplets with values outside the [0.0–1.0] range were treated by scaling the triplet so that the maximum value was 1.0 and by clipping negative values to 0.0. Finally, sRGB values were obtained by applying gamma correction with gamma = 2.2 and rescaling to the [0–255] range.

**Synthetic cells fabrication**. The synthetic cells were fabricated using a resist lift-off process. We spin coated the positive resist AZ1505 (Microchemicals GmbH) on a silicon wafer at 3000 RPM. The wafer was then baked at 100 °C for 60 s and flood exposed with a 300 mJ cm$^{-2}$ dose of UV light. A layer of the negative tone resist SU-8 2000.5 (Kayaku Advanced Materials, Inc.) was then spin coated on the positive resist layer at 3000 RPM and baked at 100 °C for 60 s. Using a photolitography mask with 30 μm square openings we exposed the SU-8 to a 60 mJ cm$^{-2}$ dose. The wafer was then baked at 95 °C for 60 s. The SU-8 layer was subsequently developed using SU-8 developer (Kayaku Advanced Materials, Inc.) for 60 s with gentle agitation. Finally, the wafer was submersed in MIF AZ 726 developer (Microchemicals GmbH) to dissolve the AZ1505 layer and obtain a solution with suspended SU-8 synthetic cells.

**Cell culture and solution preparation**. HCT-116 cells were cultured in a high glucose GlutaMAX DMEM media (Thermo Fisher Scientific) supplemented with 10% fetal bovine serum and incubated at 37 °C with 5% $CO_2$. Solutions containing cells were

prepared by washing the cultivated cells with warm Dulbecco's Phosphate Buffered Saline (Sigma-Aldrich), and fixing them with 4% paraformaldehyde (Sigma-Aldrich).

**Reporting summary**. Further information on research design is available in the Nature Portfolio Reporting Summary linked to this article.

## Data availability
All data supporting the findings of this study is available via the repository: https://github.com/burguea/biocolors.

## Code availability
All custom algorithms used in this study available at the following GitHub repository: https://github.com/burguea/biocolors.

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

## Author contributions
A.B.L., M. B., B.N.M.d.O., F.G, Y.T., V.M., and N.L. developed and characterized the Pd Surface. A.B.L., F.G., and A.F. built the optical setup. B.N.M.d.O., and A.G. prepared the HCT-116 cells. A.B.L. and F.G. fabricated the synthetic cells. M.B., Y.T, V.M., N.L., and A.F. conceived the analysis of microorganisms using thin-film interference and performed preliminary experiments. A.B.L., F.G., and B.N.M.d.O. imaged the cells on the Pd. A.B.L. and M.M. wrote the code and performed the data analysis. C.L. and A.F. supervised the project. A.B.L. wrote the paper with editorial inputs from M.M., C.L., and A.F.

## Competing interests
The authors declare no competing interests.
