## [Peer Review File · Communications Biology]

Reviewers' comments:

Reviewer #1 (Remarks to the Author):

In this manuscript, the authors present a label-free imaging technique that combines nanostructured membranes and machine learning to accurately map the refractive index (RI) and thickness of biological specimens. The approach enables sub-cellular segmentation of colorectal cancer cells and holds potential for real-time, high-resolution imaging in biomedical studies, offering valuable insights for research and medical diagnostics.

The paper could be publishable after addressing the points raised below:

- The authors should provide more details about the nanostructured membrane, including information on its engineering process and the specific properties that enable accurate RI and thickness mapping.
- The authors should elaborate on how they validated the accuracy of the machine learning-based technique for RI and thickness recovery. Additionally, discussing any limitations or challenges encountered during the validation process could be provided.
- The authors should provide a more detailed explanation of how the technique was applied in the specific context of sub-cellular segmentation of colorectal cancer cells, along with the insights gained from the segmentation results.

The authors should discuss their vision regarding the contribution of this label-free segmentation technique to biomedical studies on microscopic multicellular dynamics, particularly highlighting specific areas of research or medical diagnostics that could benefit the most from this technology in the Conclusion and Discussion section.

- The authors may address the key factors that need to be addressed or improved upon to achieve real-time capabilities using this technique for biomedical studies.
- The authors should compare their technique to other existing methods for RI mapping, such as plasmonic probe scanning or quantitative phase microscopy, in terms of accuracy, speed, and ease of implementation.
- The authors should provide more information about the training process for the machine learning model, including the type of data used for training and the model's robustness in handling different types of biological specimens or variations in imaging conditions.

Reviewer #2 (Remarks to the Author):

The manuscript titled "Label-free segmentation of sub-cellular biology at the diffraction limit in real-time" describes a technique to simultaneously measure thickness and refractive index of thin films. This is done by stretching them over a surface with broadband low reflectivity and applying machine learning algorithms to extract the thickness and RI of the film from the spectrum of the reflected light (converted to RGB values by a color camera). The authors show that their approach works on a synthetic cell (squares of cured photoresist) and then show its application to the dehydrated human cancer cell.

While color of the thin film is widely used to characterize its thickness in fabrication, I have not been able to find references where both thickness and refractive index are extracted from the color. Therefore I find the approach interesting. However, the presentation is not very convincing so far. I can not recommend the manuscript for publication unless the following questions are addressed by the authors:

- 1) Do authors include dispersion in their models? In the SI they mention that they assume a constant RI for PMMA from 400 nm to 700 nm, but in reality it seems to vary from 1.508 to 1.489 (10.1364/AO.383831), much larger than the claimed RI sensitivity of $1e-4$. How do those variations affect the calculations? Can the plots shown in Supplementary Fig. 2d be fit with theory? I am not sure what is the expected dispersion for dehydrated cellular material in that wavelength window, but I think

authors need to comment on that.

2) I do not find the example with a synthetic cell particularly convincing. First it is unclear what exactly is shown in Supplementary figure 1. What does measurement path refer to? Second the relevant regions of Fig 5c and 5d (i.e the cell itself) are correspondingly at the lower and at the upper range of the color scale, therefore it is impossible to see any structure. From Fig. 5a the "cell" does not seem to be at all uniform in thickness, as is implied by Fig 5c. It is also unclear how the uncertainty presented in Fig. 5e and 5f is calculated. I am not sure why authors refer to it as relative uncertainty in the text, as it seems to be absolute (the color scale is in nm). It would be helpful if the authors showed 2D profilometer measurement together with a 2D thickness map on a scale which can show some height variations (or the line cuts of the profilometer, as in the Supplementary figure 1, together with the corresponding line cuts of the extracted thickness). I also think since the real cell is a very complex structure, a more complex structure is necessary as a validation, e.g. polystyrene spheres, as was done in reference 24, or a sample with some kind of surface thickness variation, that can be verified using AFM or profilometry independently.

3) It is somewhat unclear what is the purpose of K-means clustering of pixels. Can't the thickness and RI be extracted for each pixel individually?

4) For the real cell measurements I do not understand how a sharp edge (Fig. 6c) is extracted from a blurry image (Fig. 6a). Moreover I am confused why the thickness of the region outside of the synthetic cell (fig 5c) is larger than that of the synthetic cell, while the thickness of the region outside of the real cell (Fig. 6c) is smaller than that of the real cell. I am also not sure how the cell thickness is estimated from the SEM image (fig. 6e) or even if it is the same cell. Finally, the typical RIs of various organelles in a cell vary between 1.36 and 1.6 (10.1039/c5lc01445j) for the cells suspended in culture medium. It is surprising to see the extracted RI to be between 1.33 and 1.6 for dehydrated cells. It would expect them to be larger. Can authors comment on that? It is also unclear why various organelles (e.g. nucleus, which typically has lower refractive index than cytosol) can not be distinguished in a cell or whether the clusters shown in Fig. 6d are corresponding to different organelles. Dehydration clearly must affect the cell properties. Can authors explain why aren't measurements performed in PBS? Is that because the thickness of the sample needs to be below a certain value, or because the reflection at the cell-water interface would be too low? This needs to be explained in the manuscript.

Additional points:

1) Line numbers would be very helpful.

2) The first sentence of the abstract is confusing. It sounds as if composition of microorganisms and models needs to be understood.

3) Authors claim that QPM-based methods need to know the thickness of the cell to extract the RI. However, as far as I can tell at least in Ref. 24 that is not necessary.

4) The average reflectivity of the Pd surface is measured to be between 1.5 and 3%. However the reflectivity of air-cell cytosol (RI=1.36) interface is not much larger (2.3%). It is therefore unclear how the Pd surface would affect the results. Additionally the reflectivity of the pillars themselves is even larger (as one can see from Fig. 2c).

5) On the top of page 5, authors say that geometrical structures enhance the scattered thin film interference colors, however the mechanism of this enhancement is unclear.

6) In Fig 1, it is unclear what is shown in panel e. What do different colors correspond to?

7) In Fig 2. It is unclear what different color squares are in panel a. I guess its regions for panels b,c and d but it needs to be made clear in the caption.

8) In Fig. 3 it is unclear what exactly is displayed in panels b and c. Why are the plots non-uniform?

9) In Fig. 5 the figure caption does not seem to match the figure.

10) In the SI Fig 2c, are the sample numbers meaningful? Can the samples be arranged by their thickness to enable easy comparison between theory (panel e) and measurement (panel c).

11) What are the reasons for RI and thickness cutoff values in panel e? Why isn't RI extended down to 1.33 and thickness not extended between 100 and 1000 nm? I am also confused how panel e relates to measurements of thin films in main text? As far as I can see the situation in panel e is different

from that in the main text (air-material-air) vs. (air-material-silicon), but it is not made very clear. What does the lookup table look like for air-material-air situation?

Author Response to Reviews of

Label-free segmentation of sub-cellular biology at the diffraction limit in real-time

RC: Reviewer Comment, AR: Author Response, □ Manuscript text, “ ” Reference quotation

Reviewer #1

RC: In this manuscript, the authors present a label-free imaging technique that combines nanostructured membranes and machine learning to accurately map the refractive index (RI) and thickness of biological specimens. The approach enables sub-cellular segmentation of colorectal cancer cells and holds potential for real-time, high-resolution imaging in biomedical studies, offering valuable insights for research and medical diagnostics.

AR: *We thank the reviewer for their careful reading of the manuscript and their supportive statements.*

RC: The paper could be publishable after addressing the points raised below:
The authors should provide more details about the nanostructured membrane, including information on its engineering process and the specific properties that enable accurate RI and thickness mapping.

AR: *We expanded the manuscript with lines 85-88, describing the engineering of the Pd membrane and its properties. We also provided additional details on the engineering of the membrane in the methods section. An excerpt of the new text follows:*

We optimize the deposition potential and time to create large and prominent tree-like feature (Fig 2a, black area) and achieve broadband light absorption. The combination of the Pd surface texture and its low reflectivity produce the cell stretching to thin film effect while simultaneously allowing the thin film interference colors to be detectable.

RC: The authors should elaborate on how they validated the accuracy of the machine learning-based technique for RI and thickness recovery. Additionally, discussing any limitations or challenges encountered during the validation process could be provided.

AR: *We expanded on the description of the validation of the technique with the synthetic cells, including comments on the challenge of imaging the cells with a low depth of field objective through lines 176-182 of the manuscript:*

We measured the cell's thickness t using optical and contact profilometry (see Supplementary Figure 1), obtaining $t = (567 \pm 6)$ nm, and obtained the ground truth RI from the resist manufacturer datasheet. Fig 5a shows a photograph of a synthetic cell through a reflection microscope at $100\times$ magnification. The blurring on the right side of Fig 5a does not originate from a thickness variation but is the result of a slight tilt of the cell, which places this area outside the depth of field of the $100\times$, 0.9 NA, objective we use to acquire the image.

RC: The authors should provide a more detailed explanation of how the technique was applied in the specific context of sub-cellular segmentation of colorectal cancer cells, along with the insights gained from the segmentation results.

AR: *We expanded on the manuscript description of the colorectal cancer cell results, including insights regarding the refractive index values obtained during the segmentation process and previously reported results through lines 195-206 of the text:*

Fig 6 presents the results of the recovery process applied to a natural cell. Fig 6a shows a photograph of an HCT-116 colon cancer cell after deposition and stretching on the Pd surface. Spatially varying thin film interference colors are visible across the specimen. The dark spots in the central part of the cell correspond to debris from a Pd pillar that moved over the cell during the deposition process. The blurriness on edge results from the short depth of field of the 100 \times , 0.9 NA objective used to capture the image. Figures Fig 6b-c show the ML computed RI and thickness maps of the specimen using 50 color clusters. This number results in a maximum variation considering all clusters of 1.98% between the RGB values of the pixels and their cluster centroid RGB triplet. The RI values shown in Fig 6b are larger than those reported in the literature for living HCT-116 cells by approximately 0.1 RIU [29, 30]. This RI increase is a consequence of cell dehydration. It is consistent with the previously reported RI increase of up to 0.15 RIU across the visible wavelength range for dehydrated tissues and isolated cells undergoing dehydration [31,32].

RC: The authors should discuss their vision regarding the contribution of this label-free segmentation technique to biomedical studies on microscopic multicellular dynamics, particularly highlighting specific areas of research or medical diagnostics that could benefit the most from this technology in the Conclusion and Discussion section.

AR: *We expanded on the manuscript discussion section through lines 253-266 as shown below, adding the additional reference:*
– Duval, K. et al. *Modeling Physiological Events in 2D vs. 3D Cell Culture*. *Physiology* **32**, 266–277 (July 2017)
and examples on how this technique can contribute to improving the understating of multicellular ensembles, with specific examples pertaining to medical diagnosis.

In the cancer care community, this method could help by automatically screening refractive index markers in cells with simple, integrated equipment with massive deployment. Differentiating between healthy and cancerous cells can be achieved with a RI resolution of 10^{-2} RIU [25,37], which we demonstrate in this work.

Future research could use the method introduced here to study in-vitro cell culture sheets. Placing cellular monolayers on the Pd substrate would enable the single measurement study of cellular growth, differentiation, and responses to biophysical or biochemical cues through the lens of RI mapping for multicellular assemblies[1,38]. The technology can also help the detection of other refractive index markers indicating changing diseases in the human blood, such as malaria, with a resolution of 10^{-2} RIU in the RI map of erythrocytes[7]. The result of this work can also benefit pharmacological research. It could provide low-cost and easy-to-use equipment that monitors the effect of new antibiotics from changes in the refractive index of bacterially infected cells, a task that requires a 10^{-3} RIU resolution[39], attainable with this technology.

RC: The authors may address the key factors that need to be addressed or improved upon to achieve real-time capabilities using this technique for biomedical studies.

AR: *We expanded the discussion section to comment on paths for improvement of the technique's real-time capabilities in lines 246-250:*

While the cell measurement is instantaneous in this technique, combining this measuring platform with automated pipetting systems could enhance the setup speed by accelerating the specimen deposition process. Instruments that accelerate the dehydration process by positioning the substrate in a dehydration chamber would further reduce the setup to measurement time, providing options for developing commercial cytometry equipment based on this platform.

RC: The authors should compare their technique to other existing methods for RI mapping, such as plasmonic probe scanning or quantitative phase microscopy, in terms of accuracy, speed, and ease of implementation.

AR: *We expanded the comparison with other state-of-the-art techniques in lines 43-49:*

Currently, state-of-the-art RI mapping compromises between accuracy and speed. The most accurate RI mapping method, plasmonic probe scanning, provides RI measurements with 10^{-5} RIU resolution, but requires tens of minutes to scan a single line of a sample's RI map [20]. Single-shot quantitative phase microscopy (QPM) provides faster estimations but requires pre-existing knowledge of the cell thickness [21]. However, the unavoidable uncertainty in estimating the cell geometry is a source of significant errors in QPM-based RI estimation, leading to measurements that vary between 10^{-2} RIU and 10^{-4} RIU in resolution [22–25].

RC: The authors should provide more information about the training process for the machine learning model, including the type of data used for training and the model's robustness in handling different types of biological specimens or variations in imaging conditions.

AR: *We provided additional detail on the manuscript through lines 128-139 and in the supplementary material concerning the training process and the robustness of the model:*

The process starts by accurately characterizing the camera's CMFs through supervised learning. In this step, we used a training and validation experimental dataset of 65 thin films of known thickness and RI. We manufactured these thin films via the spin coating of PMMA photoresist on silicon wafer pieces at different speeds and measured their thickness and RI through spectroscopic ellipsometry (See Suppl. Fig. 1). We then acquired reflection spectra and photograph pairs for each film sample. Using these samples, we trained a regression model using non-linear basis functions (see Supplementary Sec "Camera characterization" for implementation details). This approach yields the CMFs up to the desired resolution in frequency, controlled by the size of the regression model. This training process allows the measurement of any biological thin film imaged by the camera, as the ML algorithm is agnostic to the type of cell or imaged material, learning only the relation between the spectral power distribution of the specimen and the color outputted by the camera.

Reviewer #2

RC: The manuscript titled “Label-free segmentation of sub-cellular biology at the diffraction limit in real-time” describes a technique to simultaneously measure thickness and refractive index of thin films. This is done by stretching them over a surface with broadband low reflectivity and applying machine learning algorithms to extract the thickness and RI of the film from the spectrum of the reflected light (converted to RGB values by a color camera). The authors show that their approach works on a synthetic cell (squares of cured photoresist) and then show its application to the dehydrated human cancer cell. While color of the thin film is widely used to characterize its thickness in fabrication, I have not been able to find references where both thickness and refractive index are extracted from the color. Therefore I find the approach interesting

AR: *We thank the reviewer for their careful reading of the manuscript and positive evaluation of our work.*

RC: However, the presentation is not very convincing so far. I can not recommend the manuscript for publication unless the following questions are addressed by the authors: Do authors include dispersion in their models? In the SI they mention that they assume a constant RI for PMMA from 400 nm to 700 nm, but in reality it seems to vary from 1.508 to 1.489 (10.1364/AO.383831), much larger than the claimed RI sensitivity of 10^{-4} . How do those variations affect the calculations?

AR: *We revised the the SI on lines 81-87 and clarified that the dispersive variation in the PMMA RI from 400 nm to 700 nm is smaller than the 10^{-2} sensitivity we can achieve with the 12 bit camera used in this work. As such, there is no need to include this variation in the computation, so we use the average refractive index value:*

While PMMA is dispersive from 400 nm to 700 nm, the refractive index variation is smaller than the 10^{-2} sensitivity we can achieve with the 12 bit camera used in this work (See Fig. 3 in the main text) [1]. We therefore consider the average value of the PMMA refractive index in this range for our computations.

RC: Can the plots shown in Supplementary Fig. 2d be fit with theory?

AR: *We followed the Referee suggestion and assembled a new Supplementary Figure 2, reproduced in this document as Fig. A1. We have included plots of the theoretical reflection spectra for the samples shown in the old Supplementary Fig. 2d as panel d of this new figure.*

RC: I am not sure what is the expected dispersion for dehydrated cellular material in that wavelength window, but I think authors need to comment on that.

AR: *We acted on the Referee’s comment and clarified the manuscript, commenting on the fact that the refractive index increases across this wavelength window as a consequence of the dehydration process on lines 203-206:*

The RI values shown in Fig 6b are larger than those reported in the literature for living HCT-116 cells by approximately 0.1 RIU [29,30]. This RI increase is a consequence of cell dehydration. It is consistent with the previously reported RI increase of up to 0.15 RIU across the visible wavelength range for dehydrated tissues and isolated cells undergoing dehydration [31,32].

RC: I do not find the example with a synthetic cell particularly convincing. First it is unclear what exactly is shown in Supplementary figure 1. What does measurement path refer to?

Figure A1: **New supplementary Figure 2:a.** The measuring setup consists of a reflection microscope setup with Köhler illumination. An additional beam splitter is used in conjunction with a spectrum analyzer to capture the reflection spectra of the calibration samples but is not required for the cell analysis. Panel **b** shows a photograph of the calibration samples, these consist of PMMA photoresist spin coated to different thicknesses on silicon wafer pieces. Panel **c** shows the measured thickness for all calibration samples, the color of each bar corresponds to the average color of the sample. Panel **d** shows examples of the images seen through the setup for three calibration samples, alongside their measured and theoretical reflection spectra. Panel **e** shows the lookup table of thickness and refractive index to RGB for PMMA thin films seen by the camera. Panel **f** shows the lookup table of thickness and refractive index to RGB for biological thin films seen by the camera.

AR: *We have revised the paper and added a new supplementary Fig. 1, reported here as Fig. A2 for convenience. In the caption of this figure, we clarified that panel b shows a profilometer measurement of a series of synthetic cells in a row and that the measurement path is the distance traveled by the profilometer cantilever.*

RC: *Second, the relevant regions of Fig 5c and 5d (i.e the cell itself) are correspondingly at the lower and at the upper range of the color scale, therefore it is impossible to see any structure.*

AR: *We clarified the manuscript to indicate we do not expect to see any structure due to the synthetic cell's uniformity in both thickness and refractive index through lines 185-188:*

Figures 5c-d show the ML calculated thickness and RI maps of the artificial cell structure. As the cell is uniform in both thickness and refractive index, the plots present constant values for both quantities over the surface of the cell, save for the areas where the Pd pillars are detected.

Figure A2: **New supplementary Figure 1.** **a** Two dimensional optical profilometer thickness measurement of a group of synthetic cells prior to the removal of the AZ1505 supporting resist layer. **b** 1D profilometer measurement of the cells shown in panel **a**. The x axis of the plot corresponds to the distance travelled by the profilometer cantilever over the wafer surface, while the y axis shows the recorded height.

RC: From Fig. 5a the “cell” does not seem to be at all uniform in thickness, as is implied by Fig 5c.

AR: *We revised the manuscript and explained through lines 179-182 that the apparent disuniformity originates from blurring artifacts unrelated to thickness variations in the artificial cell, which is uniform by fabrication:*

Fig. 5a shows a photograph of a synthetic cell through a reflection microscope at $100\times$ magnification. The blurring on the right side of Fig. 5a does not originate from a thickness variation but is the result of a slight tilt of the cell, which places this area outside the depth of field of the $100\times$, 0.9 NA, objective we use to acquire the image.

RC: It is also unclear how the uncertainty presented in Fig. 5e and 5f is calculated. I am not sure why authors refer to it as relative uncertainty in the text, as it seems to be absolute (the color scale is in nm).

AR: *We acted on the Referee’s remark and have revised the wording “relative uncertainty” to “absolute uncertainty”. In addition, we clarified the calculation procedure for the uncertainty in the manuscript by adding lines 190-192:*

We calculated the uncertainty as the difference between the values recovered by our algorithm and ground truth measurements of the refractive index and thickness.

RC: It would be helpful if the authors showed 2D profilometer measurement together with a 2D thickness map on a scale which can show some height variations (or the line cuts of the profilometer, as in the Supplementary figure 1, together with the corresponding line cuts of the extracted thickness).

AR: *We followed the Referee’s suggestions and performed an additional 2D profilometer measurement on the synthetic cells. We presented the results of this measurement as panel **a** of the new Supplementary Figure 1, reproduced in this document as Fig. A2. This new figure shows the 2D thickness map of the synthetic cells alongside the line cuts of measured thickness. This calibration demonstrates the ability of our system to point-to-point map the required refractive index and thickness with diffraction limited resolution.*

RC: It is somewhat unclear what is the purpose of K-means clustering of pixels. Can't the thickness and RI be extracted for each pixel individually?

AR: *We revised the text and clarified between lines 140-145 that while the algorithm can extract thickness and RI for each pixel, the K-means clustering process helps overcoming metamerism:*

After estimating the CMFs, the ML recovery algorithm can extract the thickness and RI values for each pixel of a sample's image. However, due to metamerism, working with each pixel as an isolated element can result in incorrect recoveries. The ML algorithm addresses this by pooling information from pixels with close RGB values, generating groups of adjacent pixels possessing similar RGB colors in the image. This process uses an unsupervised k-means clustering algorithm that labels pixels of similar RGB colors as belonging to the same cluster.

RC: For the real cell measurements I do not understand how a sharp edge (Fig. 6c) is extracted from a blurry image (Fig. 6a).

AR: *We improved the manuscript and explained between lines 206-210 that our clustering algorithm automatically identifies the sharp edge from the RGB threshold value used to distinguish color pixels belonging to the cell from the background:*

The ML algorithm correctly isolates the Pd background in both results, grouping all pixels with low RGB values into the background cluster. This clustering step produces a sharp boundary separating the cell from the Pd according to whether the RGB values of the pixels are above the threshold the algorithm defines as the background.

RC: Moreover I am confused why the thickness of the region outside of the synthetic cell (fig 5c) is larger than that of the synthetic cell, while the thickness of the region outside of the real cell (Fig. 6c) is smaller than that of the real cell.

AR: *We revised the text in lines 188-190 and clarified that the algorithm treats the of the regions outside the cell as black thin films with no further processing:*

Our algorithm treats the Pd pillars background as a black thin film during the calculations, and will not further processes these areas for RI and thickness recovery.

RC: I am also not sure how the cell thickness is estimated from the SEM image (fig. 6e) or even if it is the same cell.

AR: *We acted on the Referee's comment and improved the manuscript by clarifying the measuring process in lines 216-220:*

We ensured the SEM imaged cell was the same as the cell shown in Figure 6a by scratching markings in the Pd surrounding the cell. We estimated the cell thickness from the SEM image by measuring the number of pixels in the image corresponding to the raised border of the cell, and then multiplying this value by the size in nanometers of one pixel.

RC: Finally, the typical RIs of various organelles in a cell vary between 1.36 and 1.6 (10.1039/c5lc01445j) for the cells suspended in culture medium. It is surprising to see the extracted RI to be between 1.33 and 1.6 for dehydrated cells. It would expect them to be larger. Can authors comment on that?

AR: *We have followed the Referee's comment and revised the text between lines 203-206, as shown below, clarifying that recovered RI values are larger as a consequence of dehydration, which is known to increase the refractive index. We have added the following references as part of the revision:*

– Sun, L. et al. *Graphene-Based Confocal Refractive Index Microscopy for Label-Free Differentiation of Living Epithelial and Mesenchymal Cells*. ACS Sensors **5**, 510–518 (Feb. 28, 2020)

– Sun, L. et al. *Refractive Index Mapping of Single Cells with a Graphene-Based Optical Sensor*. Sensors and Actuators B: Chemical **242**, 41–46 (Apr. 1, 2017)

– Oliveira, L. et al. *Optical Characterization and Composition of Abdominal Wall Muscle from Rat*. Optics and Lasers in Engineering **47**, 667–672 (June 1, 2009)

– Beuthan, J. et al. *The Spatial Variation of the Refractive Index in Biological Cells*. Physics in Medicine & Biology **41**, 369 (Mar. 1996)

The RI values shown in Fig 6b are larger than those reported in the literature for living HCT-116 cells by approximately 0.1 RIU [29,30]. This RI increase is a consequence of cell dehydration. It is consistent with the previously reported RI increase of up to 0.15 RIU across the visible wavelength range for dehydrated tissues and isolated cells undergoing dehydration [31,32].

RC: Can authors explain why aren't measurements performed in PBS? Is that because the thickness of the sample needs to be below a certain value, or because the reflection at the cell-water interface would be too low? This needs to be explained in the manuscript.

AR: *We have revised the manuscript in lines 63-66 by clarifying that the measurements cannot be performed in PBS because the sample has to dehydrate to stretch to sub-micron thickness:*

The hydrophilic nature of the Pd surface causes the PBS to spread over the sample, resulting in the evaporation of the liquid within one minute of the deposition. The lack of liquid produces progressive dehydration of the specimen, causing it to flatten and stretch on the surface, forming a suspended, thin biological film.

RC: Additional points:
Line numbers would be very helpful.

AR: *We have added line numbers to the main text and supplementary information files.*

RC: The first sentence of the abstract is confusing. It sounds as if composition of microorganisms and models needs to be understood.

AR: *We followed the Referee's suggestion and revised the abstract through lines 13-15:*

Mapping the cellular refractive index (RI) is a central task for research involving the composition of microorganisms and the development of models providing automated medical screenings with accuracy beyond 95%.

RC: Authors claim that QPM-based methods need to know the thickness of the cell to extract the RI. However, as far as I can tell at least in Ref. 24 that is not necessary.

AR: *We improved the manuscript in lines 46-47 and clarified that we refer to the family of single-shot methods, to which our technique belongs.*

Single-shot quantitative phase microscopy (QPM) provides faster estimations but requires pre-existing knowledge of the cell thickness [21]

RC: The average reflectivity of the Pd surface is measured to be between 1.5 and 3%. However the reflectivity of air-cell cytosol (RI=1.36) interface is not much larger (2.3%). It is therefore unclear how the Pd surface would affect the results. Additionally the reflectivity of the pillars themselves is even larger (as one can see from Fig. 2c).

AR: *We acted on the Referee's question and revised the main text in lines 95-99 as shown below, explaining that the reflectivity of the Pillars in Fig. 2c results from an integrating sphere measurement, which considers all the scattered components of light. Because of the strongly nanostructured nature of the pillars, scattering occurs at high angles. Thin film interference colors, on the contrary, have mostly scattering components within the numerical aperture of the microscope objective. As such, they are detectable, as we clearly show in our images.*

The inset in Fig. 2c reports the region's reflectivity across visible wavelengths measured with an integrating sphere, showing that the nanostructured Pd reflects less than 2% of visible light relative to a silver mirror. Most of the light scattering from the pillars occurs at high angles, enabling the detection of thin film interference components that scatter within the numerical aperture of the microscope objective.

RC: On the top of page 5, authors say that geometrical structures enhance the scattered thin film interference colors, however the mechanism of this enhancement is unclear.

AR: *We have added lines 86-88 to clarify this point:*

The combination of the Pd surface texture and its low reflectivity produce the cell stretching to thin film effect while simultaneously allowing the thin film interference colors to be detectable.

RC: In Fig 1, it is unclear what is shown in panel e. What do different colors correspond to?

AR: *We expanded the caption of Fig 1 to clarify that the color overlays indicate subcellular regions with similar refractive index:*

e Micrographs of an HCT-116 cell. The color overlays indicate subcellular regions with similar refractive index.

RC: In Fig 2. It is unclear what different color squares are in panel a. I guess its regions for panels b,c and d but it needs to be made clear in the caption.

AR: We expanded the caption of Fig 2 to clarify that the squares correspond to the regions shown in panels b, c and d:

a. Photograph of nanostructured Pd sample. The color squares correspond to the regions imaged in panels **b, c** and **d**.

RC: In Fig. 3 it is unclear what exactly is displayed in panels b and c. Why are the plots non-uniform?

AR: We have revised the panels and caption of Fig. 3 to clarify that panels **b** and **c** show the sensitivity limits for our technique as a function of the camera's bit depth. The shape of the plots shown in panels **b** and **c** is due to the discrete nature of the y axis, which has units of bits. We have modified the y axis of the graphs in **b** and **c** to display integer numbers that better reflect this. We reproduce the modified figure in this document as A3

Figure A3: **New figure 3.** **a.** Biological thin film colors in the sRGB colorspace for four different thicknesses as the refractive index varies from 1.33 to 1.55. **b, c** Sensitivity limits for refractive index and thickness values recovery as a function of the channel bit depth of the camera used (colored lines) and the stability of the image values (y axis). The y axis is in discrete units of bits, while the x axis is continuous.

RC: In Fig. 5 the figure caption does not seem to match the figure.

AR: We have amended the figure caption as requested:

a. Photograph of a synthetic cell as seen under $100\times$ magnification on top of the Pd substrate. The two dark spots correspond to Pd pillars visible through the cell. **b.** 3D model showing the relative positioning of the synthetic cell on the Pd pillars. **c-d.** 3D reconstruction of the thickness and refractive index maps obtained for the synthetic cell. **e-f.** Uncertainty maps for the thickness and RI of the synthetic cell.

RC: In the SI Fig 2c, are the sample numbers meaningful? Can the samples be arranged by their thickness to enable easy comparison between theory (panel e) and measurement (panel c).

AR: *We have plotted the calibration samples ordered by thickness as panel c of a new Supplementary Figure 2. Reproduced in this text as Fig. A1*

RC: What are the reasons for RI and thickness cutoff values in panel e? Why isn't RI extended down to 1.33 and thickness not extended between 100 and 1000 nm?

AR: *We have revised the figure and performed additional calculations expanding Fig. 2e's limits to encompass the full range of RI and thickness values of the calibration samples. We reproduce the new panel in this document as element e of Fig. A1.*

RC: I am also confused how panel e relates to measurements of thin films in main text? As far as I can see the situation in panel e is different from that in the main text (air-material-air) vs. (air-material-silicon), but it is not made very clear. What does the lookup table look like for air-material-air situation?

AR: *We have improved the manuscript and added a lookup table for air-material-air to the figure as panel f of the new Supplementary Figure 2, and have modified the caption to clarify the difference between the two lookup tables. We reproduce the image here as panel f of Fig. A1.*

Reviewers' comments:

Reviewer #1 (Remarks to the Author):

The authors addressed all of my concerns. I agree for the publication.

Reviewer #2 (Remarks to the Author):

In my second review of the manuscript "Label-free segmentation of sub-cellular biology at the diffraction limit in real time", I found that the clarity of the presentation has been significantly improved. I would now like the authors to address the following points.

First, while the authors do extract the thickness and refractive index of different parts of the dehydrated cell, I find the term "segmentation of subcellular biology" misleading. So I am not sure if the title is appropriate. Also, something as large as the nucleus, which should have a lower RI than the cytoplasm (10.1002/jbio.201500273) and should be visible with any technique, is missing from the images generated by the discussed method. Can the authors comment on why this is the case?

The artificial cell example is now clearer: all pixels have been grouped by the ML algorithm and therefore have the same thickness and RI extracted. And it looks like the variation in RGB values between neighboring pixels is less than 2%? Why was the 2% threshold chosen? Is it related to the sensitivity of the measurement? The error in thickness recovery is reported to be 0.6 nm. However, in Figure S1, the variation in thickness between different cells and within individual cells appears to be significantly larger than 0.6 nm. Can you comment on this?

While I understand that ML clustering helps when metameric samples are present (as explained in Figure 4), is it possible to extract the thickness and RI for each individual pixel after clustering? For example, in Figure 4c, we see that pixel 2 was originally identified as having a different RI and thickness. After clustering, it is combined with the rest. Can the maps shown in Figure 4c now be used to extract the thickness and RI for each pixel, conditional on it belonging to a particular group? If this were applied to the synthetic cell in Figure 5, the thickness of the RI maps would not necessarily be uniform and would more accurately represent the true thickness?

Could you also discuss what are the thickness limits of the objects that can be measured with this technique? In the manuscript, all objects measured were thinner than 1 μm . Does the thickness/RI extraction become more complicated for thicker samples? Can the measurement range be extended by using a light source with a longer coherence length?

Could you also discuss how sensitive the extracted thickness and RI maps are to the position of the focal plane?

A few other minor points:

Figure 3b-c is still unclear. If the y-axis is discrete, why is each plot continuous and composed of three separate segments?

For example, for a 14-bit channel, why is the minimum n independent of the maximum bit change as long as the maximum bit change is greater than 1? How is the maximum bit change measured for a given setup?

Also, in Figure S1, I still don't see how panels a and b relate to each other. Can you show the measurement path in panel a?

Figure 2: Very difficult to read the labels and ticks on the inset of c. I would also suggest swapping the

position of a and c.

Figure 4 d,e: Fonts have different sizes.

Figure 6 bcf: Fonts have different sizes, in f the color bar is labeled "refractive index", in b "refractive index".

Line 132: See Figure S2

Author Response to Reviews of

Real-time simultaneous refractive index and thickness mapping of sub-cellular biology at the diffraction limit

RC: Reviewer Comment, AR: Author Response, □ Manuscript text, “ ” Reference quotation

Reviewer #1

RC: The authors addressed all of my concerns. I agree for the publication.

AR: *We thank the reviewer for their help in improving the manuscript and endorsement towards publication.*

Reviewer #2

RC: In my second review of the manuscript “Label-free segmentation of sub-cellular biology at the diffraction limit in real time”, I found that the clarity of the presentation has been significantly improved. I would now like the authors to address the following points.

AR: *We thank the reviewer for the positive comment on the work.*

RC: First, while the authors do extract the thickness and refractive index of different parts of the dehydrated cell, I find the term "segmentation of subcellular biology" misleading. So I am not sure if the title is appropriate.

AR: *We followed the reviewer’s suggestion and changed the title to: “Real-time simultaneous refractive index and thickness mapping of sub-cellular biology at the diffraction limit”.*

RC: Also, something as large as the nucleus, which should have a lower RI than the cytoplasm (10.1002/jbio.201500273) and should be visible with any technique, is missing from the images generated by the discussed method. Can the authors comment on why this is the case?

AR: *We clarified in the revised work through lines 212-214, reproduced below, that in the case of refractive index mapping for HCT-116 cells, the cytoplasm-nucleus boundary is not readily apparent as shown in the following articles.*

- Sun, L. et al. *Graphene-Based Confocal Refractive Index Microscopy for Label-Free Differentiation of Living Epithelial and Mesenchymal Cells*. ACS Sensors **5**, 510–518 (Feb. 28, 2020)
- Sun, L. et al. *Refractive Index Mapping of Single Cells with a Graphene-Based Optical Sensor*. Sensors and Actuators B: Chemical **242**, 41–46 (Apr. 1, 2017)

Consistently with previously reported RI maps for HCT-116 cells, no sharp nucleous-cytoplasm boundary is apparent, however, the RI values shown in Fig. 6b are larger than those reported in the literature for living HCT-116 cells by approximately 0.1 RIU [29,30]

RC: The artificial cell example is now clearer: all pixels have been grouped by the ML algorithm and therefore have the same thickness and RI extracted. And it looks like the variation in RGB values between neighboring pixels is less than 2%? Why was the 2% threshold chosen? Is it related to the sensitivity of the measurement?

AR: *We acted on the Referee question and clarified the work by adding lines 145-153 to the manuscript:*

The ML recovery procedure automatically sets the number of clusters to yield an average variation of less than 2% between the RGB values of the pixels in each cluster and the cluster centroid RGB value. We set this value as a threshold found through successive iterations of the algorithm, with the condition that a lower value would result in the differences in recovered RI and thickness values for the pixels in a cluster being below the sensitivity of our setup.

RC: The error in thickness recovery is reported to be 0.6 nm. However, in Figure S1, the variation in thickness between different cells and within individual cells appears to be significantly larger than 0.6 nm. Can you comment on this?

AR: *We improved the text by adding lines 196-201, which clarify the calculation is with respect to the average cell thickness of the profilometer measurements.*

We calculated the uncertainty as the difference between the values recovered by our algorithm and ground truth measurements of the refractive index and thickness. The procedure yields results with an average discrepancy of 0.6 nm in the thickness recovery compared to the average cell thickness obtained with the profilometer measurements and of 3×10^{-3} RIU compared to the datasheet RI over the synthetic cell area.

RC: While I understand that ML clustering helps when metameric samples are present (as explained in Figure 4), is it possible to extract the thickness and RI for each individual pixel after clustering? For example, in Figure 4c, we see that pixel 2 was originally identified as having a different RI and thickness. After clustering, it is combined with the rest. Can the maps shown in Figure 4c now be used to extract the thickness and RI for each pixel, conditional on it belonging to a particular group? If this were applied to the synthetic cell in Figure 5, the thickness of the RI maps would not necessarily be uniform and would more accurately represent the true thickness?

AR: *Yes, it is possible to extract the thickness and RI of each pixel in a cluster after the clustering step. However, in the case of Fig. 5, the synthetic cell is uniform in thickness and RI by design, so any pixel-wise extraction would provide data with differences below the sensitivity limit of the experimental setup, thus not providing information.*

RC: Could you also discuss what are the thickness limits of the objects that can be measured with this technique? In the manuscript, all objects measured were thinner than 1 μm . Does the thickness/RI extraction become more complicated for thicker samples? Can the measurement range be extended by using a light source with a longer coherence length?

AR: *We revised the text and added lines 96-103 in the supplementary information:*

This technique requires the appearance of thin film interference colors from the sample. The visibility of the colors decreases when the sample's thickness is significantly larger than the wavelength. Figs. 2 e-f show this trend on the far right of the plots. Increasing the measurement range requires increasing the source's coherence length while ensuring a sufficiently broad spectral coverage.

RC: Could you also discuss how sensitive the extracted thickness and RI maps are to the position of the focal plane?

AR: *We addressed this question and improved the manuscript by adding lines 207-209:*

We set the microscope to focus on the largest possible cell area as the sample must be in focus to prevent overlap between neighboring pixels' RGB values and allow the technique to obtain sharp RI and thickness maps

RC: A few other minor points:

RC: Figure 3b-c is still unclear. If the y-axis is discrete, why is each plot continuous and composed of three separate segments? For example, for a 14-bit channel, why is the minimum n independent of the maximum bit change as long as the maximum bit change is greater than 1? How is the maximum bit change measured for a given setup?

AR: *Following the reviewer's comment, we have re-plotted Fig. 3b-c, and modified its description to better reflect the discrete nature of the y-axis. We also added lines 108-116 to the manuscript that expand on the figure description. We reproduce the new text below and the new figure as Fig. A1 of this document*

The y-axis of the plots represents the level of variation, in units of bits, that the image file may suffer from due to thermal, electrical, or illumination fluctuations in the experimental setup. This value can be estimated by examining the variation in pixel values between images of the same object taken at different times. For a given bit variation, each circle marks the thickness or refractive index resolution below which two distinct biological structures yield the same RGB triplet. The dotted lines of the image help visualizing the resolution dependence on bit depth, but the plots are not continuous as a discrete variation of camera bit depth yields a discrete variation in the sensitivity of the technique.

RC: Also, in Figure S1, I still don't see how panels a and b relate to each other. Can you show the measurement path in panel a?

AR: *We revised figure S1 with an additional panel showing the measurement path equivalent for panel a, and clarified the figure shows measurements with different profilometers. We reproduce the revised figure here as Fig. A2*

RC: Figure 2: Very difficult to read the labels and ticks on the inset of c. I would also suggest swapping the position of a and c.

AR: *We acted on the reviewer's comment, modified the figure with a larger font size, and swapped the panels. We reproduce it here as Fig. A3*

RC: Figure 4 d,e: Fonts have different sizes.

Figure A1: Revised figure 3 of the main manuscript

AR: We revised the panels to have the same font size. We reproduce the modified figure here as Fig. A4

RC: Figure 6 bcf: Fonts have different sizes, in f the color bar is labeled “Refractive index”, in b “refractive index”.

AR: We revised the font size of the panels and made the capitalization uniform across the figure. We reproduce the revised figure here as Fig. A5

RC: Line 132: See Figure S2

AR: We revised all references in the main manuscript to supplementary material to comply with the style and formatting guide of Nature communications biology.

Figure A2: Revised figure 1 of the supplementary text

Figure A3: Revised figure 2 of the main manuscript

Figure A4: Revised figure 4 of the main manuscript

Figure A5: Revised figure 6 of the main manuscript

Reviewer #2 (Remarks to the Author):

In my third revision of this manuscript, I find that my criticisms have been adequately addressed. I believe that the manuscript could be improved further by including more rigorous analyses for known samples of different thicknesses and different RIs. Nevertheless, it could be published in its present form.

Author Response to Reviews of

Real-time simultaneous refractive index and thickness mapping of sub-cellular biology at the diffraction limit

RC: Reviewer Comment, AR: *Author Response*, □ Manuscript text, “ ” Reference quotation

Reviewer #2

RC: In my third revision of this manuscript, I find that my criticisms have been adequately addressed. I believe that the manuscript could be improved further by including more rigorous analyses for known samples of different thicknesses and different RIs. Nevertheless, it could be published in its present form.

AR: *We thank the reviewer for their help in improving the manuscript and endorsement towards publication.*